# Cell-Penetrating Peptide Modified PEG-PLA Micelles for Efficient PTX Delivery

**DOI:** 10.3390/ijms21051856

**Published:** 2020-03-09

**Authors:** Qi Shuai, Yue Cai, Guangkuo Zhao, Xuanrong Sun

**Affiliations:** Collaborative Innovation Center of Yangtze River Delta Region Green Pharmaceuticals, Zhejiang University of Technology, Hangzhou 310006, China; qshuai@zjut.edu.cn (Q.S.); caiyue1992@foxmail.com (Y.C.); zgk7721@163.com (G.Z.)

**Keywords:** cell-penetrating peptide, nanoparticles, active tumor targeting, PEG-PLA

## Abstract

On account of their excellent capacity to significantly improve the bioavailability and solubility of chemotherapy drugs, amphiphilic block copolymer-based micelles have been widely utilized for chemotherapy drug delivery. In order to further improve the antitumor ability and to also reduce undesired side effects of drugs, cell-penetrating peptides have been used to functionalize the surface of polymer micelles endowed with the ability to target tumor tissues. Herein, we first synthesized functional polyethylene glycol-polylactic acid (PEG-PLA) tethered with maleimide at the PEG section of the block polymer, which was further conjugated with a specific peptide, the transactivating transcriptional activator (TAT), with an approved capacity of aiding translocation across the plasma membrane. Then, TAT-conjugated, paclitaxel-loaded nanoparticles were self-assembled into stable nanoparticles with a favorable size of 20 nm, and displayed a significantly increased cytotoxicity, due to their enhanced accumulation via peptide-mediated cellular association in human breast cancer cells (MCF-7) *in vitro*. But when further used *in vivo*, TAT-NP-PTX showed an acceleration of the drug’s plasma clearance rate compared with NP-PTX, and therefore weakened its antitumor activities in the mice model, because of its positive charge, its elimination by the endoplasmic reticulum system more quickly, and its targeting effect on normal cells leading towards being more toxic. So further modification of TAT-NP-PTX to shield TAT peptide’s positive charges may be a hot topic to overcome the present dilemma.

## 1. Introduction

Cancer imposes a huge burden upon modern society [1]. Chemotherapy is usually considered as the most common choice for cancer treatment. As one of the most important anticancer chemotherapy drugs, Paclitaxel (PTX) has been widely used for the treatment of a wide variety of cancers, including breast, lung and advanced ovarian cancers [2,3]. However, the therapeutic efficacy of PTX has been impeded by its intrinsic rapid clearance, poor water solubility and high toxicity. In addition, a nonselective biodistribution of PTX results in the poor accumulation of PTX at the tumor sites, which leads to inefficient tumor suppression and serious side effects. For the purpose of improving its pharmacokinetics and realizing more aggressive tumor infiltration, the strategy of fabricating polymeric PTX-loaded nanoparticles has been widely utilized, and a large number of successful cases have been presented [4,5,6,7]. Synthetic amphiphilic PEG-PLA block-polymer and its various functional derivatives have been extensively applied for nanomedicine, and both PEG and PLA have been approved for clinical applications by the United States Food and Drug Administration (FDA) [8,9]. Nontoxic, biodegradable PEG-PLA has been widely used to encapsulate PTX and form water-soluble nanoparticles, achieving improved water solubility and bioavailability of PTX [10,11]. Meanwhile, PEG-PLA can be easily prepared by conventional ring-opening polymerization, and the modification of both ends of the block polymer chain is feasible for different purposes. All of these features of PEG-PLA make it a potential candidate for drug delivery.

It is commonly acknowledged that due to increased tumor vascular permeability and nonfunctional lymphatic drainage, chemotherapy drugs-loaded nanoparticles can passively accumulate in the tumor microenvironment by an enhanced permeability and retention (EPR) effect [12]. Nevertheless, increased interstitial fluid pressure, as well as tumor heterogeneity and stroma, can seriously impair the efficacy of passively targeted drugs, thus most of these nanoparticles are only confined to the perivascular area, and cannot effectively penetrate deep into tumor tissues or cells far from blood vessels [13,14]. In order to achieve efficient active tumor targeting, and thus inhibit tumor growth with the fewest adverse effects, cell-penetrating peptides (CPPs) have been explored to functionalize the surface of polymer micelles [15]. CPPs are small hydrophilic peptides that can transport larger molecules into cells without relying on classical endocytosis [16].

The transactivating transcriptional activator (TAT) peptide, a cell-penetrating peptide with a specific amino acid sequence (YGRKKRRQRRRC) [17,18,19], which has the special ability to aid translocation across the plasma membrane [20], is widely used to modify the anticancer drug carriers, and indeed improve the drug accumulation deep in tumor tissues [21,22,23].

Herein, the TAT peptide was chosen to conjugate with maleimide-PEG-PLA (Mal-PEG-PLA) via a maleimide–thiol coupling reaction [24]. The resulting TAT-conjugated PEG-PLA (TAT-PEG-PLA) was used to prepare PTX-loaded nanoparticles (TAT-NP-PTX). Both *in-vitro* tumor targeting, penetration capacity and *in-vivo* antitumor efficiency, pharmacokinetics and organ biodistribution on breast cancer cells, were systemically evaluated for these novel nanoparticles. 

## 2. Results

### 2.1. Synthesis of Mal-PEG-PLA

Maleimide-terminated polyethylene glycol-polylactic acid (PEG-PLA) were prepared according to the process illustrated in Appendix A. Firstly, the carboxylation of PEG with excess succinic anhydride obtained a mixture of monocarboxyl (HOOC-PEG-OH) and dicarboxyl-modified PEG (HOOC-PEG-COOH). However, it was difficult to remove undesired HOOC-PEG-COOH by column chromatography because of the close polarities of the two polymers. Instead, the product mixture was directly used for the subsequent polymerization step, since HOOC-PEG-COOH was inert under the ring-open polymerization conditions. As a result, a monocarboxyl-terminated block-polymer HOOC-PEG-PLA was obtained in the presence of unreacted HOOC-PEG-COOH, which was able to be removed by hot water washing [25]. Finally, HEMI was connected to HOOC-PEG-PLA via an esterification reaction, affording Mal-PEG-PLA.

Appendix A displays the protonic nuclear magnetic resonance (^1^H-NMR) spectrum of the chemical composition of the synthesized products. The spectrum (Appendix A) shows the presence of the maleimide group (δ = 6.7 ppm) at the PEG terminus. As shown in Appendix A, the Mn of the PLA block in the HOOC-PEG-PLA increased from 1460 Da to 1930 Da after hot water washing, indicating that the HOOC-PEG-COOH has been removed. Moreover, Appendix A likewise displays a typical gel permeation chromatography (GPC) trajectory of the purified copolymer, which has further confirmed the remove of HOOC-PEG-COOH. The Mn of finally synthesized Mal-PEG-PLA was about 4100 Da. 

### 2.2. Characterization of PTX Loaded Nanoparticles

The preparation of nanoparticles is mentioned in the supporting information. 

The critical micelle concentration (CMC) is a critical property for micelles, as it indicates the capacity of polymer to form micelles in aqueous media. As shown in Figure 1A, the CMC of PEG-PLA nanoparticles was around 9.25 μg/mL, and that of TAT-PEG-PLA nanoparticles (Figure 1B) was around 14.29 μg/mL, which was increased due to a decrease of the hydrophobic/hydrophilic segment ratio [26].

The results of the dynamic light scattering (DLS) test displayed that the average particle size of NP-PTX was about 19 nm, with a narrow size distribution (Table 1), which might permit more efficient penetration in tumor tissues, due to the enhanced permeability and retention (EPR) effect [27]. After modification with TAT peptides, the average size of TAT-NP-PTX nanoparticles was increased to 20 nm. The charge of NP-PTX was approximately -6.34 mv, while that of TAT-NP-PTX was significantly increased to +5.94 mv (Table 1). The obvious inversion of zeta potential strongly confirmed the successful conjugation of positive charge TAT peptides onto the surface of NP-PTX. Representative transmission electron microscopy (TEM) morphology identified that NP-PTX and TAT-NP-PTX exhibited a uniform regular spherical shape (Figure 1C,D).

The drug loading rate (LR%) of the NP-PTX and TAT -NP-PTX was 1.24% and 1.10% respectively, while the entrapment efficiency (EE%) was 62.1% and 55.3%, respectively.

The stability of NP-PTX and TAT-NP-PTX was observed in fetal bovine serum (FBS) and phosphate-buffered saline (PBS), respectively, at the concentration of 1 mg/mL, and the change of the average particle size was measured by dynamic light scattering (DLS) within five days. As is shown in Figure 1E, both NP-PTX and TAT-NP-PTX exhibited a satisfactory stability under different physiological environments. The diameter of TAT-NP-PTX in FBS increased slightly from 37.7 nm to 53.7 nm, while it decreased from 33.2 nm to 23.9 nm in PBS. For NP-PTX, only a small fluctuation of the diameter change both in FBS and PBS was observed.

### 2.3. In Vitro PTX Release Profiles

*In-vitro* PTX release behavior was tested in PBS with 0.1% of Tween-80 at pH 7.4 and pH 5.0 respectively, in order to mimic the physiologic and tumor microenvironment. As shown in Figure 1F, generally, the release behaviors of NP-PTX and TAT-NP-PTX *in vitro* were quite similar. After 96 h incubation in PBS (pH 7.4), the cumulative release of PTX from NP-PTX and TAT-NP-PTX was 77.93% ± 3.03% and 77.67% ± 1.69%, respectively. 

However, PTX was released more quickly from the nanoparticles over the same period of time at pH 5.0 (90.90% ± 2.56% for NP-PTX and 88.33% ± 1.25% for TAT-NP-PTX). The faster release of PTX under acidic conditions might be contributed to the promoted hydrolysis of PEG-PLA under acid conditions. These results implied that TAT-NP-PTX could potentially release PTX rapidly once penetrating into the tumor microenvironment. 

### 2.4. In Vitro Cytotoxicity Studies

The Thiazolyl Blue Tetrazolium Bromide (MTT) method was utilized to evaluate the cytotoxicity of NP-PTX, TAT-NP-PTX and free Taxol on MCF-7 cells. As indicated in Figure 2A, TAT-NP-PTX revealed much higher tumor cell growth inhibition (IC50 values = 2.4 ng/mL on MCF-7 cells) than that of NP-PTX (IC50 = 17.6 ng/mL) and Taxol (IC50 = 31.6 ng/mL), which might due to a much more effective accumulation of TAT-NP-PTX on tumor cells and the positive charge of the nanoparticles. As we expected, blank nanoparticles (PEG-PLA nanoparticles without PTX) showed almost nontoxicity on tumor cells, which identified the safety of the polymeric carriers. 

### 2.5. Inhibition Ability on MCF-7 Tumor Spheroid

The *in-vitro* cell cytotoxicity of NP-PTX and TAT-NP-PTX was further evaluated by investigating their suppression ability on the growth of tumor spheroid, which could better mimic the microenvironment of the *in-vivo* tumor. As shown in Figure 2B,C, the tumor spheroid incubated with Dulbecco’s Modified Eagle Medium (DMEM) grew up at a relatively high rate, with the spheroid volume reaching to four times larger after 9 days incubation. Obviously, when treated with Taxol, NP-PTX, TAT-NP-PTX at the equivalent of PTX concentration of 10 ng/mL, significant growth inhibition of the tumor spheroids was observed within 9 days. Especially, the group which treated with TAT-NP-PTX had the minimum tumor spheroids growth rate compared with other PTX formulations, indicating that TAT-NP-PTX had the best tumor-penetrating ability and antitumor activity as we expected, indicating its better tumor-penetrating ability and potential better antitumor effect *in vivo*.

### 2.6. Penetration of Nanoparticles into Tumor Spheroids

In order to test the penetration capability of TAT-NP-PTX into tumor spheroids, the MCF-7 3D tumor spheroids model was established as mentioned above. As shown in Figure 2D, fluorescence was mainly observed on the edge of the sphere treated with the nanoparticles which were labeled by coumarin 6 (NP-C6, 0.23‰ C6 loading rate), while a much higher intensity of fluorescence was detected in multicellular tumor spheroids when treated with TAT peptide modified, coumarin 6-labeled nanoparticles (TAT-NP-C6, 0.21‰ C6 loading rate), revealing a better penetration capability of TAT-NP-C6, as we expected. It demonstrated that TAT-NP-PTX modified with TAT was endowed with a much better penetration ability through tumor tissues than regular NP-PTX. 

### 2.7. Uptake and Cellular Localization

In the cause of demonstrating the efficacy of TAT in mediating the cellular uptake of nanoparticles, a cellular localization of nanoparticles labeled by Coumarin 6 on MCF-7 cells was performed. ImageJ was used to quantify the fluorescence intensity by choosing 15 cells in each image. As shown in Figure 3A,B, MCF-7 cells treated by TAT-NP-C6 displayed much stronger fluorescence intensity than that exposed to NP-C6, and the fluorescence intensity was concentration-dependent. 

Quantitative results of flow cytometry were in agreement with the consequences of qualitative analysis. As shown in Figure 3C, the cellular uptakes of NP-C6 and TAT-NP-C6 by MCF-7 cells were both concentration-dependent. It was also observed that, compared with NP-C6, TAT-NP-C6 was more rapidly ingested by MCF-7 cells due to the presence of TAT on the surface of corresponding nanoparticles. 

Intracellular localization of nanoparticles was shown in Figure 3D. After incubation with C6-loaded nanoparticles for 2 h, intensive green fluorescence could be seen in the cytoplasm of MCF-7 cells, clearly indicating that both NP and TAT-NP were located in cytoplasm after internalization. 

### 2.8. In Vivo Pharmacokinetics and Biodistribution

As shown in Figure 4A and Table 2, Taxol was rapidly cleared from the blood, which meant that only a small amount of free Taxol would reach the tumor tissue. Compared with Taxol, the blood circulation time of PTX encapsulated in both NP-PTX and TAT-NP-PTX was significantly extended. However, NP-PTX was superior in improvement to TAT-NP-PTX, due to its positive surface charge. The good performance of these nanoparticles in extending the circulation time of PTX was attributed to pegylation effect which imparted stealth shields to the nanoparticles [28,29]. However, TAT-NP-PTX did not display the same significant improvement as NP -PTX, due to its positive surface charge accelerating the drug’s plasma clearance rate to some extent. 

The *in-vivo* biodistribution and targeting capacity of TAT-NP-PTX were both determined on MCF-7 tumor-bearing mice. As shown in Figure 4B, both NP-PTX and TAT-PTX displayed decreased accumulation in liver and spleen compared to Taxol, consistent with the higher plasma concentration results. It also showed that there was significantly higher PTX accumulation in tumor tissue for TAT-NP-PTX, about 2-fold of that for Taxol, which indicates once reaching to the tumor tissue, where TAT-NP-PTX could penetrate more deeper into the tumor. However, there was no significant difference in the accumulation of these two kinds of nanoparticles in the tumor. 

Above all, TAT-NP-PTX had longer blood circulation time and better tumor-targeting ability than Taxol, but the installation of the TAT peptide to the nanoparticles did not show a much-enhanced accumulation of PTX at the tumor location as we desired. If the positive surface charge of TAT-NP-PTX could be shielded before entering the tumor tissue, the blood circulation time and tumor-targeting ability of TAT-NP-PTX may be greatly improved.

### 2.9. In Vivo Antitumor Activity and Safety Studies

Due to its excellent *in-vitro* performance on tumor cells, the antitumor ability of TAT-NP-PTX on MCF-7 tumor-bearing nude mice were then evaluated. It was obvious that all of Taxol, NP-PTX and TAT-NP-PTX could suppress the growth of MCF-7 tumors compared with the PBS control group, and the tumor inhibition rates were 31.9%, 39.9%, 24.0%, respectively (Figure 5A). 

To further confirm this result, the tumor weights were determined after dissection. As shown in Figure 5B, the tumor weights of mice treated with Taxol, NP-PTX and TAT-NP-PTX were 1.5 ± 0.14 g, 1.19 ± 0.06 g and 1.36 ± 0.25 g, respectively. The change of tumor weights was basically consistent with the change in tumor volume. However, compared with NP-PTX, TAT-NP-PTX did not show superior antitumor ability *in vivo*, as it did within *in-vitro* experiments, which was out of our expectation.

Hematoxylin-eosin (H&E) staining and terminal deoxynucleotidyl transferase nick end labeling (TUNEL) staining were conducted to perform histological analysis and further confirm the antitumor activity of nanoparticles. As shown in Figure 5D, no apoptosis was observed for tumors treated with PBS, whereas treatment with NP-PTX and TAT-NP-PTX resulted in serious apoptosis.

TUNEL staining identified that a portion of MCF-7 tumor cells with apoptotic brown staining were observed in the groups treated with NP-PTX and TAT-NP-PTX, indicating *in-vivo* antitumor efficiency of these nanoparticles. However, the results of histological analysis still had not revealed the ideal antitumor activity of TAT-NP-PTX *in vivo*.

Moreover, as shown in Figure 5C, 12 days after injection, mice treated with TAT-NP-PTX showed weight loss. Meanwhile, during the treatment, it was found that the tails of mice treated with TAT-NP-PTX showed slight injuries from the fifth day after the injection. The body weight loss and the damage of tails may be ascribed to a strong, positive charge of TAT-NP-PTX.

With these results in hand, it was concluded that TAT-NP-PTX did not show better tumor accumulation in the *in-vivo* distribution experiments compared with NP-PTX, however its antitumor effect was weakened by its greater toxicity to normal cells. Furthermore, H&E staining was also assessed to study the biosafety of nanoparticles, as shown in Figure 6. No visible damage of all the major organs (heart, liver spleen, lung, kidney) in the H&E staining was observed.

## 3. Discussion

Herein, we firstly successfully prepared pure carboxyl-terminated amphiphilic block polymer HOOC-PEG-PLA in an easy and convenient way, avoiding a difficult column chromatography purification process and reducing many unnecessary, repetitive tasks. Then, HOOC-PEG-PLA was further functionalized with a thiol-active maleimide moiety, making it readily reactive to specific peptides. In the next step, a novel PTX delivery system for PTX with tumor-penetrating capability was established on TAT peptides-modified PEG-PLA. Compared with Taxol and NP-PTX, the obtained TAT-NP-PTX with size of 20.18 nm and inverted zeta potential of +5.94 mv showed the best antitumor effect *in vitro*, due to its enhanced accumulation via a peptide-mediated cellular association in MCF-7 cells, indicating its great potential to inhibit tumor. However, the *in-vivo* antitumor effect of TAT-NP-PTX was not as good as we expected. TAT-NP-PTX might be eliminated by the endoplasmic reticulum system because of its positive charge, which could accelerate the plasma clearance rate of the drug. Meanwhile, unselective targeting of TAT peptides towards normal cells made the nanoparticles somewhat toxic, and further impaired their *in-vivo* antitumor effect.

In summary, it is true that TAT-NP-PTX has shown strong tumor penetration and antitumor ability *in vitro*, but further modification of TAT-NP-PTX will be necessary to shield TAT peptides before the nanoparticles enter tumor tissues and expose the TAT peptide in the tumor microenvironment to prolong the blood circulation time and enhance the antitumor ability *in vivo*. Further modification and application of this delivery system is undergoing in our laboratory. 

## 4. Materials and Methods

### 4.1. Materials

The transactivating transcriptional activator (TAT) peptide (YGRKKRRQRRRC) with purity ≥97% was purchased from ChinaPeptides Co., Ltd. (Shanghai, China). Taxol injection was obtained from Yangtze River Pharmaceutical (Group) Co., Ltd. (Jiangsu, China). Hochest33342 and 4% paraformaldehyde were provided by Solarbio Science and Technology Co., Ltd. (Beijing, China). Acetonitrile for high-performance liquid chromatography (HPLC) was obtained from Tedia Company, Inc (Fairfeld, OH, USA). Anhydrous pyridine, 1-(2-Hydroxyethyl)-1H-pyrrole-2,5-dione (HEMI), PEG (Mw2000) and D, L-lactide were obtained from Aladdin Bio-Chem Technology Co., LTD (Shanghai, China). Dulbecco’s Modified Eagle Medium (DMEM) (high glucose) cell culture medium, penicillin/streptomycin stock solutions, fetal bovine serum (FBS) and Trypsin were all bought from Gibco BRL (Gaithersberg, MD, USA). Protonic nuclear magnetic resonance (^1^H-NMR) (400 MHz) was used to confirm the obtained compounds (Bruker AVVANCE DRX-400 NMR spectrometer, Bruker, Switzerland). In addition, gel permeation chromatography (GPC) (Waters 1515–2414, Waters, USA) was used to analyze all polymers by using tetrahydrofuran as the mobile phase (flow rate: 1 mL/min) at 30 °C. Polystyrene was used as standards.

### 4.2. Cells and Animals

MCF-7 cells were purchased from the China Center for Type Culture Collection (Wuhan, China) and cultured in DMEM containing 10% FBS and 1% antibiotics (penicillin/streptomycin).

6-week-old female ICR mice (18–22 g of body weight) and BALB/c mice (4–5 weeks old, 14–16 g of body weight) were obtained from Shanghai Slac Laboratory Animal Co. Ltd. All animal procedures were approved by the Animal Experimental Center of Zhejiang University of Technology (Hangzhou, China) and followed the guidelines for care and use of laboratory animals.

### 4.3. Measurement of the Critical Micelle Concentration (CMC)

A solution of Nile (0.5 µg) in CH_2_Cl_2_ (15 µL) was added to several light-proof glass bottles, respectively, and the CH_2_Cl_2_ was then evaporated by vacuum pump [30]. Then, TAT-PEG-PLA and PEG-PLA aqueous solutions (1.5 mL) with series concentrations ranging from 0.1 µg/mL to 100 μg/mL were added into the bottles with stirring. After 12 h of stirring, the fluorescence of Nile red (Emission wavelength: 525 nm; excitation wavelength 485 nm) was determined by using a microplate reader (Flexstation 3, Molecular Devices LLC, Sunnyvale, CA, USA).

### 4.4. Characterization of NP-PTX and TAT-NP-PTX

The particle size and surface zeta potential of the nanoparticles were measured utilizing a dynamic light scattering (DLS) detector (Zetasizer, Nano-ZS90, Malvern, UK) at 25 °C. The morphological observation of the nanoparticles was performed using a JEM-1200EX transmission electron microscope (Jeol Ltd., Japan) following staining with 2% uranyl acetate. The stabilities of these nanoparticles were further tested in FBS and PBS (pH 7.4) within 5 days.

High performance liquid chromatography (HPLC) was used to determine the entrapment efficiency (EE%) and drug loading rate (LR%). Briefly, 1 mg of nanoparticles was dissolved in 1 mL acetonitrile, and the resulting solution was then analyzed on an Agilent 1260 HPLC system (Agilent, USA), using the reversed-phase column (Inertsil C18, 5 μm, 4.6 mm × 250 mm, Japan) at room temperature. Acetonitrile and ultra-pure water (60:40 *v*/*v*) were used as the mobile phase, the flow rate was 1.0 mL/min, and the detection wavelength was 227 nm. The EE% and LE% were figured as below:(1)$$\mathrm{EE}\left(\%\right)=\frac{\mathrm{PTX}\;\mathrm{content}\;\mathrm{in}\;\mathrm{the}\;\mathrm{nanoparticles}}{\mathrm{Total}\;\mathrm{amount}\;\mathrm{of}\;\mathrm{PTX}\;\mathrm{added}}\times 100\%$$
(2)$$\mathrm{LR}\left(\%\right)=\frac{\mathrm{PTX}\;\mathrm{content}\;\mathrm{in}\;\mathrm{the}\;\mathrm{nanoparticles}}{\mathrm{Nanoparticles}\prime \;\mathrm{weight}}\times 100\%$$

### 4.5. In Vitro PTX Release

The *in-vitro* release behavior of PTX was determined by a dialysis method under different pH conditions, as reported before [31]. In brief, 1 mL of NP-PTX or TAT-NP-PTX formulation containing 0.1 mg PTX was loaded into a dialysis bag (MWCO = 3500 Da), which was then immersed in 29 mL of phosphate buffer solution (PBS, pH 7.4 and 5.0, respectively) with 0.1% *w*/*v* Tween-80 with stirring at 100 rpm for 96 h at 37 °C. At particular time intervals in time, 0.2 mL release medium was taken out for a test, and 0.2 mL of fresh corresponding medium was added. The PTX levels of those samples was measured by HPLC using acetonitrile and ultra-pure water (60:40 *v*/*v*) as a mobile phase (flow rate: 1 mL/min; detection wavelength: 227 nm). 

### 4.6. In vitro Cytotoxicity Studies

MCF-7 cells were incubated in 96-well plates at the density of 5 × 10^3^ cells per well. After 12 h incubation, the medium was replaced with 200 µL fresh medium containing Taxol injection, NP-PTX or TAT-NP-PTX (at the PTX concentrations of 0.01 µg/mL, 0.1 µg/mL, 1 µg/mL, 5 µg/mL, 10 µg/mL). After another 72 h incubation, 20 µL MTT solution with the concentration of 5 mg/mL was added to each well. 4 h later, the medium was removed and 100 µL of dimethyl sulfoxide (DMSO) was added to each well. The absorbance at 570 nm of each well was measured using a microplate reader (Flexstation 3, Molecular Devices LLC, Sunnyvale, CA, USA).

### 4.7. Cellular Localization and Uptake of Coumarin-6-Loaded Nanoparticles in Breast Cancer Cells

For qualitative analysis and intracellular localization, MCF-7 cells were seeded in a 24-well plate at the density of 2 × 10^4^ cells per well. After 12 h incubation, the cells were treated with coumarin-6-loaded NPs (NP-C6) and TAT peptide-conjugated coumarin-6-loaded nanoparticles (TAT-NP-C6) at the nanoparticle concentrations ranging from 50 µg/mL to 400 µg/mL. Two hours later, the cells were washed with 4 °C PBS twice, fixed with 4% paraformaldehyde solution for 15 min at ambient temperature, and then stained with Hochest33342 for another 10 min. After being washed with 4 °C PBS twice, the plate was observed under a fluorescent microscope (Olympus IX73, Japan) and the results were analyzed by ImageJ software. 

For quantitative analysis, MCF-7 cells were incubated in a 6-well plate at the density of 1 × 10^6^ cells per well. 24 h later, the cells were exposed to NP-C6 and TAT-NP-C6 at the nanoparticle concentrations ranging from 50 µg/mL to 400 µg/mL for 2 h. Then the cells were trypsinized after washing with 4 °C PBS for two times. After centrifugation in cold PBS at 1000 rpm for 5 min, the cells were washed with 4 °C PBS twice. Finally, the cells were resuspended in 4 °C PBS, and analyzed by a flow cytometry (Beckman, USA) 

### 4.8. Inhibition Ability on Tumor Spheroid

MCF-7 tumor spheroid models were established as below [32]. In brief, 50 µL ultrapure water with 1.5% agarose (*w*/*v*) was added to each well of a 96-well plate. MCF-7 cells were then diluted to a density around 1 × 10^5^ cells/mL in DMEM containing 0.24% (*w*/*v*) methylcellulose. 20 μL of the above-mentioned medium were dropped on the lid of a 10 cm^2^ cell culture dish. 24 h later, the spheroids were taken out and transferred to the prearranged 96-well plate coated with agarose. One spheroid was incubated in one well. After another 72-h incubation, multicellular tumor spheroids about 400 μm in diameter were obtained. Subsequently, TAT-NP-PTX, NP-PTX and Taxol at the equivalent of 10 ng/mL PTX concentration were added to the 96-well plate, and the tumor spheroid diameter in each well was determined by using an inverted phase microscope (TS100-F, Nikon, Japan) for nine days. The maximum diameter (d_max_) and minimum diameter (d_min_) of every spheroid were surveyed, and the volume was calculated by utilizing the equation: V = (Π × d_max_ × d_min_)/6. The negative control was the spheroids incubated in DMEM medium.

### 4.9. Penetration Ability on Tumor Spheroid

The MCF-7 tumor spheroids were prepared as mentioned above. The penetration capacity of the nanoparticles was evaluated by treating spheroids with NP-C6 and TAT-NP-C6 at a nanoparticle concentration of 400 μg/mL. After 4 h incubation, the tumor spheroids were fixed by 4% paraformaldehyde, and the level of permeability of these nanoparticles was analyzed by a laser scanning confocal microscopy at the excitation wavelength about 450 nm (CLSM, Nikon-A1, Japan).

### 4.10. Pharmacokinetic Evaluation and in vivo Tissue Biodistribution Studies

Female ICR mice (≈20 g) were intravenously administrated with Taxol injection, NP-PTX and TAT-NP-PTX (an equivalent PTX dose of 7.5 mg/kg) via the tail vein. At appropriate time intervals, 0.2 mL of fresh blood was collected and put in heparin sodium pre-treated tubes, followed by immediate centrifugation at 5000 rpm, 4 °C for 10 min. After centrifugation, the supernatant was stored at −20 °C. Before HPLC analysis, 50 μL of the serum was mixed with 100 μL of acetonitrile to extract PTX and precipitate the proteins. The mixture was centrifuged at 12,000 rpm, 4 °C for 10 min, and the supernatant was filtered via a 220 nm filter. Finally, 20 μL of the samples was tested by HPLC using acetonitrile and ultra-pure water (55:45 *v*/*v*) as mobile phase (flow rate: 1 mL/min; detection wavelength: 227 nm).

The tumor-bearing model was established by injecting MCF-7 cells (1 × 10^7^ cells per mouse) subcutaneously in the right flank of female BALB/c mice (16–19 g) [33]. The mice were casually divided into three groups (three mice each group), and then injected with TAT-NP-PTX, NP-PTX or Taxol (10 mg PTX equiv./kg) via the tail vein when the volume of tumors increased to 200 mm^3^. The mice were sacrificed 8 h after injection. The tumor tissue, as well as major organs, including heart, liver, spleen, lung and kidney, were gathered and then homogenized with deionized water at the ratio of 1:3 (g/mL). 

Briefly, 100 µL of homogenization was mixed with 300 µL of acetonitrile, and the resulting mixture was vortexed for 1 min and centrifuged at 12,000 rpm, 4 °C for 10 min. The supernatant was gathered and dried by vacuum pump. Then 100 µL of mobile phase (acetonitrile: water = 55:45) was used to re-dissolve the residuum, and the content of PTX was analyzed by HPLC.

### 4.11. In vivo Antitumor Activity

The tumor-bearing animal model was established as mentioned above. When the volume of tumor in mice reached about 150 mm^3^ (8 days after inoculation), these mice were casually divided into four groups (six mice per group). Among the mice, three groups of mice were injected with TAT-NP-PTX, NP-PTX or Taxol every 3 days via the tail vein (PTX dosage of 5 mg/kg) for five times. Another group was given PBS as the negative control. The first day of administration was recorded as day 0, the tumor growth and body weight change were measured every 3 days, and the tumor volume was figured by using the following equation: Tumor volume = 0.5 × length × width^2^. On the fifteenth day, the animals were sacrificed, and the tumors as well as major organs (hearts, lungs, livers, kidneys and spleens) were weighed and collected for further histological examinations.

### 4.12. Statistical Analysis

Statistical analysis was expressed as the mean ± SD, using the unpaired Student’s t-test. Differences were classified as significant (* *p* < 0.05), very significant (** *p* < 0.01) and extremely significant (*** *p* < 0.001) (GraphPad Prism 7, GraphPad Software Inc., San Diego, CA, USA).

## Figures and Tables

**Figure 1 ijms-21-01856-f001:**
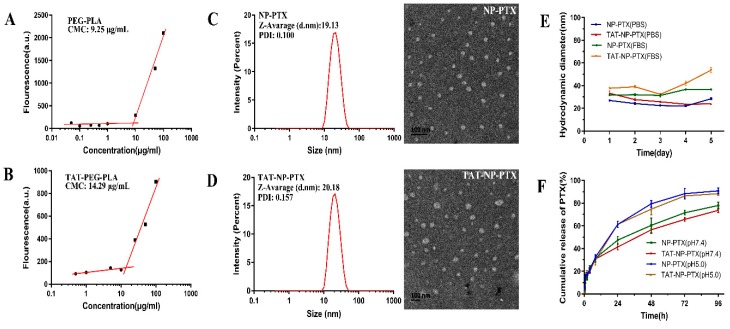
Characterization of related nanoparticles. The critical micelle concentration (CMC) value of polyethylene glycol-polylactic acid (PEG-PLA) (**A**) and TAT-PEG-PLA (**B**); Nanoparticle size and transmission electron microscopy (TEM) morphology of NP-PTX (**C**) and TAT-NP-PTX (**D**); (**E**) The stability of NP-PTX and TAT-NP-PTX in fetal bovine serum (FBS) and phosphate-buffered saline (PBS), respectively; (**F**) *In vitro* PTX release behavior of NP-PTX and TAT-NP-PTX in PBS (pH 5.0 and 7.4) containing 0.1% of Tween-80.

**Figure 2 ijms-21-01856-f002:**
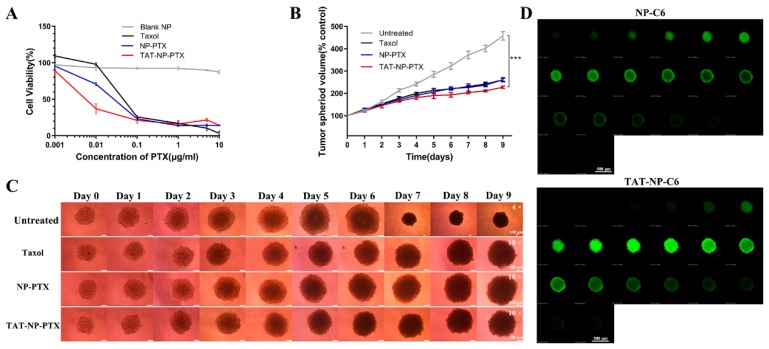
(**A**) Cell viability of MCF-7 cells after incubated with Paclitaxel (PTX) formulation for 72 h. (**B**) Quantification of tumor spheroid volume in each group based on images. Data were expressed as mean ± SD (*n* = 3, *** *p* < 0.001). (**C**) Representative bright-field images of MCF-7 tumor spheroids. (The tumor spheres of the group treated with PBS grew too fast, that they could only be observed with a smaller original magnification (4×) after day 6, while the original magnification of other images was 10×.) Bar = 100 μm. (**D**) Penetration of NP-C6 and TAT-NP-C6 in MCF-7 tumor spheroids. The spheroid was scanned from the bottom for every 20 μm. Bar = 500 μm.

**Figure 3 ijms-21-01856-f003:**
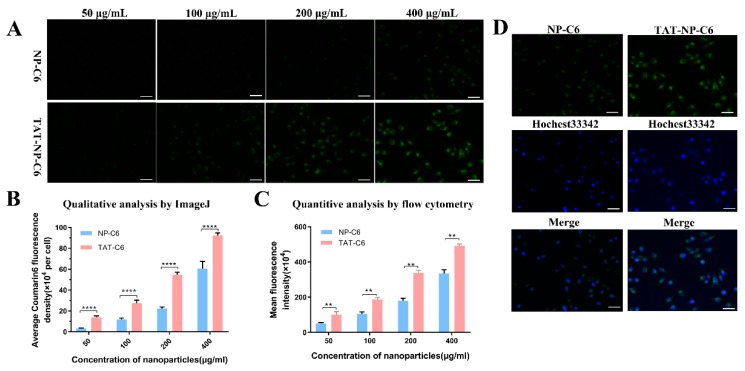
(**A**) Analysis of the cellular uptake of NP-C6 and TAT-NP-C6 in MCF-7 cells by fluorescent microscopy after 2 h incubation at different nanoparticle concentrations. Bar = 100 μm. (**B**) Qualitative results analyzed by ImageJ. Dates are expressed as mean ± SD (*n* = 3, *** *p* < 0.001). (**C**) Quantitative cellular uptake results analyzed by flow cytometry in MCF-7 cells after 2 h incubation with NP-C6 and TAT-NP-C6 at different nanoparticle concentrations. Dates are expressed as mean ± SD (*n* = 3, ** *p* < 0.01). (**D**) Intracellular localization of nanoparticles in MCF-7 cells after 2 h incubation with NP-C6 and TAT-NP-C6 at a concentration of 400 μg/mL. Bar = 100 μm.

**Figure 4 ijms-21-01856-f004:**
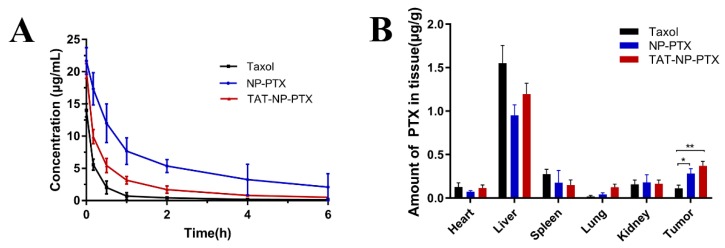
(**A**) Concentration of PTX in Institute of Cancer Research (ICR) mice plasma after tail intravenous injection of three PTX formulations at the dose of 7.5 mg/kg (*n* = 3); (**B**) Quantification of PTX accumulated in tumor tissues and major organs 8 h after injection. Dates are expressed as mean ± standard deviation (SD) (*n* = 3, * *p* < 0.05, ** *p* < 0.01).

**Figure 5 ijms-21-01856-f005:**
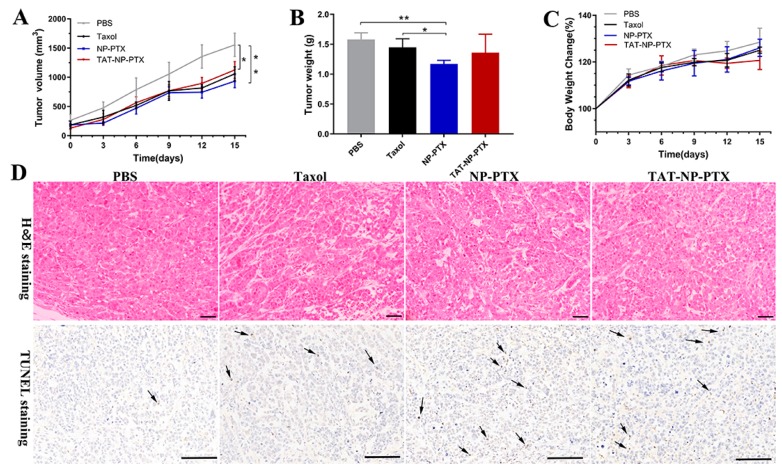
*In-vivo* antitumor effect at a dosage of 10 mg/mL PTX. (**A**) Tumor volume change curves throughout the treatment. (**B**) Average weight of tumors at the end of treatment after dissection. (**C**) Body weight change curves throughout the treatment. (**D**) Hematoxylin-eosin (H&E) and terminal deoxynucleotidyl transferase nick end labeling (TUNEL) staining of the tumor sections. Data are expressed as the mean ± SD (*n* = 6, * *p* < 0.05, ** *p* < 0.01). Bar = 100 μm.

**Figure 6 ijms-21-01856-f006:**
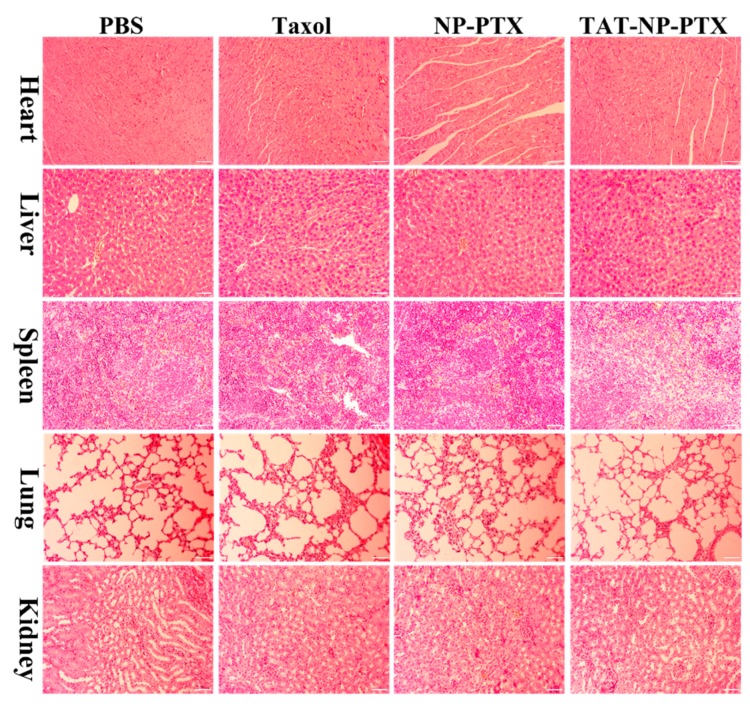
Biosafety analysis of major organs treated with PBS, Taxol, NP-PTX, TAT-NP-PTX at the equivalent PTX dose of 10 mg/kg by H&E staining. Bar = 100 μm.

**Table 1 ijms-21-01856-t001:** Characterization of NP-PTX and TAT-NP-PTX, where PTX refers to Paclitaxel, and TAT is the transactivating transcriptional activator.

Nanoparticles	Average Particle Size (nm)	Polydispersity Index (PDI)	Zeta Potential (mV)
NP-PTX	19.13	0.100	−6.34
TAT-NP-PTX	20.18	0.157	+5.94

**Table 2 ijms-21-01856-t002:** Pharmacokinetics after tail vein injection of three PTX formulations at a dose of 7.5 mg/kg in ICR mice (*n* = 3).

Parameters	Taxol	NP-PTX	TAT-NP-PTX
Cmax (mg/L)	14.019	21.667	20.10
t_1/2α_ (h)	0.029	0.193	0.064
t_1/2β_ (h)	0.192	0.412	0.344
AUC (0-∞) (mg/L*h)	5.329	42.964	14.533
CL (L/h/kg)	1.407	0.175	0.529

Abbreviations: t1/2α: distribution half-life; t1/2β: elimination half-life; AUC (0-∞): area under curve; CL: plasma clearance.

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
