# Peer review of "Cell-Penetrating Peptide Modified PEG-PLA Micelles for Efficient PTX Delivery"

_ijms, 2020, doi:10.3390/ijms21051856_

Round 1

Reviewer 1 Report

  1. Why after dialysis, the remained free PTX was removed by a 220 nm filter? Isn’t free PTX small enough to penetrate through 220 nm filter since the size of the nanoparticles containing PTX is about 20 nm?
  2. Authors should present the cytotoxicity results of nanoparticles without drug for non-tumour cells.
  3. Lines 39-41 – Authors write that synthetic amphiphilic PEG-PLA blockpolymer have been approved for clinical applications by FDA. To confirm that references or examples should be added.
  4. Authors should pay attention to superscripts and subscripts throughout the manuscript.
  5. Information about Coumarin 6 (C6) should first mention in paragraph 2.6 than 2.7.
  6. Authors should specify the concentrations of Taxol, NP-PTX or TAT-NP-PTX in 4.6 paragraph (In vitro cytotoxicity studies)

Author Response

1.Why after dialysis, the remained free PTX was removed by a 220 nm filter? Isn’t free PTX small enough to penetrate through 220 nm filter since the size of the nanoparticles containing PTX is about 20 nm?
Response: Thanks for your question. Because of the entrapment efficiency of NP-PTX and TAT-NP-PTX was 62.1% and 55.3% respectively, there were still many drug precipitations in the solution after dialysis. So, a 220 nm filter was used to removed the large aggregates of PTX. As for the very small amount of free PTX dissolved in water, it could be negligible compared to the PTX which loaded in the nanoparticles due to the poor solubility of PTX (0.48μg/mL). Besides, removing unencapsulated drugs by filtration is a common method for preparing drug-loaded nanoparticles and this method has been reported in many literatures (such as J CONTROL RELEASE 2020, 321,198-210; Nanoscale 2018, 10, 19338; Acta Pharmacologica Sinica 2017, 38, 859–873; J CONTROL RELEASE 2012, 160, 692-698).

2.Authors should present the cytotoxicity results of nanoparticles without drug for non-tumour cells.
Response: Thanks for your suggestion. The cytotoxicity of blank NP (PEG-PLA nanoparticles without PTX loaded) was evaluated in Figure 2, and there was no visible cytotoxicity. As for the TAT peptide, many researches had studied and showed no cytotoxicity (Nanoscale 2011, 3, 2315-2323; PNAS 2013, 110, 17047–17052; J CONTROL RELEASE 2015, 212 94–102). All in all, we think the above information is enough to verify the cytotoxicity of PEG-PLA and TAT-PEG-PLA.

3.Lines 39-41 – Authors write that synthetic amphiphilic PEG-PLA blockpolymer have been approved for clinical applications by FDA. To confirm that references or examples should be added.
Response: Thanks for your advice. For lines 39-41, the previous statement was not rigorous enough, so we have revised the sentence and added references.

4.Authors should pay attention to superscripts and subscripts throughout the manuscript.
Response: Thanks for your advice. We have checked and modified the article for incorrect superscript and subscript as follows: line 323, 5×103; line 332, 2×104; line 339, 1×106; line 348, 1×105; line 349, 10 cm2; line 374, 1×107.

5.Information about Coumarin 6 (C6) should first mention in paragraph 2.6 than 2.7.
Response: Thanks for your advice. We have added the information of coumarin 6 to section 2.6 (lines 153-156) and removed the previous information in section 2.7 (lines 163-166).

6.Authors should specify the concentrations of Taxol, NP-PTX or TAT-NP-PTX in 4.6 paragraph (In vitro cytotoxicity studies)
Response: Thanks for your advice. We have added the concentrations of PTX formulations used in in vitro cytotoxicity study in the lines 325-326.

Reviewer 2 Report

The overall paper is well presented and well written. Although in vivo results didn't corroborate the in vitro results, the discussion obtained is important to the development and optimizing of this nanoparticles. 

One point I think should be addressed:

Coumarin 6 (C6) should be presented first in point 2.6 of the results, instead of 2.7. 

Author Response

1. Coumarin 6 (C6) should be presented first in point 2.6 of the results, instead of 2.7.

Response: Thanks for your advice. We have added the information of coumarin 6 to section 2.6 (lines 153-156) and removed the information in section 2.7 (lines 163-166).